# Survival in Cytologically Proven Node-Positive Breast Cancer Patients with Nodal Pathological Complete Response after Neoadjuvant Chemotherapy

**DOI:** 10.3390/cancers12092633

**Published:** 2020-09-15

**Authors:** Hitoshi Inari, Natsuki Teruya, Miki Kishi, Rie Horii, Futoshi Akiyama, Shunji Takahashi, Yoshinori Ito, Takayuki Ueno, Takuji Iwase, Shinji Ohno

**Affiliations:** 1Breast Oncology Center, Cancer Institute Hospital, Japanese Foundation for Cancer Research, 3-8-31 Ariake, Koto-ku, Tokyo 135-8550, Japan; hitoshi.inari@jfcr.or.jp (H.I.); natsuki.teruya@jfcr.or.jp (N.T.); miki.kishi@jfcr.or.jp (M.K.); yito@jfcr.or.jp (Y.I.); shinji.ohno@jfcr.or.jp (S.O.); 2Department of Surgery, Saiseikai Yokohamashi Nanbu Hospital, 3-2-10 Konandai, Konan-ku, Yokohama 234-8503, Japan; 3Department of Pathology, Saitama Cancer Center, 780 Ina-machi, Saitama 362-0806, Japan; riehorii@cancer-c.pref.saitama.jp; 4Department of Pathology, Cancer Institute, Japanese Foundation for Cancer Research, 3-8-31 Ariake, Koto-ku, Tokyo 135-8550, Japan; akiyama@a-bp.net; 5Department of Medical Oncology, Cancer Institute Hospital, Japanese Foundation for Cancer Research, 3-8-31 Ariake, Koto-ku, Tokyo 135-8550, Japan; s.takahashi-chemotherapy@jfcr.or.jp; 6Breast Center, Japanese Red Cross Nagoya Daiichi Hospital, 3-35 Michishitacho, Nakamura-ku, Nagoya 453-8511, Japan; t-iwase@nagoya-1st.jrc.or.jp

**Keywords:** breast cancer, neoadjuvant chemotherapy, axillary lymph node, pathological complete response, prognosis

## Abstract

**Simple Summary:**

It is unknown whether patients with cytologically proven axillary node-positive breast cancer, who achieve axillary pathological complete response (pCR) after neoadjuvant chemotherapy (NAC) have a comparable prognosis to patients with axillary pathological node-negative disease (pN-) without NAC. The aim of this retrospective study was to clarify the clinical impact of axillary pCR after NAC on survival and to compare survival outcomes between breast cancer patients with axillary pCR, and those with axillary pN- without NAC, using propensity score matching to adjust for baseline characteristics other than nodal status. Axillary pCR after NAC was associated with improved prognosis in patients with axillary node-positive disease, and patients with axillary pCR and matched pairs with axillary pN- without NAC had comparable outcomes.

**Abstract:**

Background: It is unknown whether patients with cytologically proven axillary node-positive breast cancer who achieve axillary pathological complete response (pCR) after neoadjuvant chemotherapy (NAC) have comparable prognosis to patients with axillary pathological node-negative disease (pN-) without NAC. Methods: We retrospectively reviewed the data of patients with cytologically proven axillary node-positive disease who received NAC and those with axillary pN- without NAC for control between January 2007 and December 2012. We compared outcomes according to response in the axilla to NAC and between patients with axillary pCR and matched pairs with axillary pN- without NAC using propensity scores. Results: We included 596 patients with node-positive breast cancer who received NAC. The median follow-up period was 64 months. Patients with axillary pCR showed significantly better distant disease-free survival (DDFS) and overall survival (OS) than patients with residual axillary disease (both *p* < 0.01). There was no significant difference in DDFS and OS between patients with axillary pCR and matched pairs with axillary pN- without NAC. Conclusion: Axillary pCR was associated with improved prognosis. Patients with axillary pCR and matched pairs with axillary pN- without NAC had comparable outcomes. This information will be useful when considering the intensity of follow-up and adjuvant therapy.

## 1. Introduction

There are at least four major subtypes of breast cancer based on pathological examination: luminal, luminal-human epidermal growth factor receptor (HER)2, HER2-type, and triple negative [1]. Axillary lymph node (LN) involvement at diagnosis is considered the most critical determinant of long-term poor prognosis for patients with all subtypes of breast cancer [2,3,4,5,6,7,8]. The 20-year mortality risk for estrogen receptor (ER)-positive breast cancer is 15% for N0 disease, 28% for N1–3, and 49% for N4–9 [9]. Thus, it is critical to improve survival in patients with nodal involvement regardless of the subtype of breast cancer.

Neoadjuvant chemotherapy (NAC) is one of the standard treatment strategies for operable invasive breast cancer, especially for patients with node-positive breast cancer in all subtypes [10,11,12,13,14,15,16,17,18,19]. Various studies have demonstrated that axillary LN pathological complete response (pCR) after NAC is associated with improved prognosis in patients with clinically node-positive disease [18,19,20,21,22,23]. However, it is unknown whether patients with cytologically proven axillary node-positive breast cancer, who achieve axillary pCR after NAC have comparable prognosis to those with pathologically axillary node-negative disease who did not receive NAC. This study aimed to clarify the clinical impact of axillary pCR after NAC on survival, and to compare survival outcomes between cytologically proven node-positive breast cancer patients with axillary pCR and those with axillary pathological node-negative disease (pN-) without NAC using propensity score matching to adjust for baseline characteristics other than nodal status.

## 2. Results

### 2.1. Clinicopathological Characteristics and Survival Outcomes According to Pathological LN Response to NAC

A total of 2772 patients were enrolled (Figure 1). Of the 780 patients with cytologically proven node-positive disease, 596 received NAC. Of these, 211 achieved axillary pCR, and 385 had residual axillary disease (Figure 1). The clinicopathological characteristics of the 596 patients are shown in Table 1. T stage, nuclear grade (NG), ER and HER2 status, breast-pCR, and lymphatic invasion (LI) status differed significantly between the two groups but clinical N stage did not (Table 1).

We examined distant disease-free survival (DDFS) and overall survival (OS) according to the pathological response in the LNs (Figure 2a,b). The median follow-up time after primary surgery was 64 months (range, 9–145 months). Patients with axillary pCR showed significantly better DDFS and OS than patients with residual axillary disease (*p* < 0.01 for both DDFS and OS). The 5 year DDFS rate was 90% in patients with axillary pCR, and 70% in those with residual axillary disease. Multivariate analysis adjusted for other confounding factors related to survival outcome was performed (Table 2). Axillary pCR remained an independent prognostic factor for DDFS after adjustment for clinical stage, ER, HER2, and NG (*p* < 0.01) (Table 2).

### 2.2. Clinicopathological Factors Associated with Axillary pCR

To investigate factors associated with axillary pCR, a logistic regression analysis that includes menopausal status, cT, cN, ER, HER2 status, NG, and breast-pCR was performed (Table 3). The analysis showed that ER, HER2 status, NG, and breast-pCR showed an independent association with axillary pCR, whereas clinical T and N stages did not (Table 3).

### 2.3. Survival Outcomes between Patients with Axillary pCR and Matched Pairs with Axillary pN- without NAC Using Propensity Score Matching

To examine the clinical relevance of axillary pCR, we compared survival outcomes between patients with axillary pCR and those with axillary pN- without NAC, using propensity score matching. For the control, we enrolled patients with pT1-4 pN0 M0 breast cancer who underwent surgery without NAC (Figure 1). The variables used for propensity score matching included menopausal status, clinical T stage, ER status, HER2 status, and NG. Propensity score matching resulted in 133 patients in each group. The baseline characteristics of each group showed no significant difference between the two groups except for radiotherapy and systemic therapy (Table 4), which was expected because nodal status is an important factor for the decision of radiotherapy and systemic therapy. We compared DDFS and OS between the matched groups (Figure 3a,b). The 5 year DDFS rate was 91% for both patients with axillary pCR and those with axillary pN- without NAC. The 5 year OS rate was 95% for patients with axillary pCR, and 92% for those with axillary pN- without NAC. The log-rank test showed no significant difference in DDFS and OS between the two groups.

## 3. Discussion

Once patients with cytologically proven node-positive disease achieved axillary pCR, they showed improved DDFS and OS, compared with patients with residual axillary disease. This is in agreement with previous reports [18,19,20,21,22,23]. Axillary pCR remained prognostic for DDFS after adjustment for other prognostic factors such as clinical stage, ER, HER2, and NG. Axillary pCR after NAC can be used as an independent predictor of long-term favorable outcomes in patients with clinically node-positive disease [19].

In this study, in order to match the background characteristics other than nodal status, we used propensity score matching. After matching baseline clinicopathological characteristics between patients with axillary pCR and those with axillary pN- without NAC, we demonstrated that OS and DDFS were similar between the two groups. Patients with axillary pCR in clinically node-positive disease had a favorable prognosis, which was comparable to the prognosis of those with axillary pN- without NAC. This information will be useful to predict prognosis more practically and to decide upon the intensity of follow-up and the administration of additional therapies such as pertuzumab and capecitabine after surgery.

In this study, axillary pCR was shown to be associated with subtypes, including ER and HER2 status, high NG, and breast-response to NAC (pCR), which agrees with previous reports [19,23]. Although axillary pCR may indicate no need for axillary surgery after NAC, axillary dissection is currently the standard treatment strategy in patients with originally node-positive disease [23,24]. Accurate prediction of axillary pCR would help to avoid axillary dissection [19,23,24,25]. Application of sentinel lymph nodes biopsy (SLNB) for patients with clinically negative conversion of lymph node would be one strategy, although the false negative rate of SLNB for such patients was reported to be high at more than 10% [26,27,28,29,30]. However, consideration of factors relevant to axillary pCR, such as subtype and breast response to NAC would help in reducing the false negative rate and, consequently, to avoid unnecessary axillary dissection [25,31,32,33].

We enrolled patients who underwent surgery between 2007 and 2012, because they had a consistent strategy for systemic therapy. In particular, the chemotherapy and endocrine therapy regimens were identical to the current regimens, and the criteria for perioperative chemotherapy were similar to the current criteria, which will make the results more practically useful.

This study has several limitations. First, the treatment between the two groups was different. Baseline nodal status is an important determinant for radiotherapy and perioperative systemic therapy, especially chemotherapy, so it is understandable that some patients with node-negative disease do not need radiotherapy or chemotherapy. Nevertheless, it is clinically important that patients with axillary pCR after NAC have a prognosis comparable to those with axillary pN- without NAC, who do not necessarily need radiotherapy or chemotherapy. In order to match the background characteristics other than nodal status between the two groups, propensity score matching was used. Propensity score matching is being used with increasing frequency to account for treatment selection bias when estimating causal treatment effects using observational data [34]. We utilized this statistical method to match the background characteristics that affect prognosis except for nodal status. Another limitation was that pertuzumab was not used in patients with HER2-positive disease. Most of the patients with node-positive HER2-positive breast cancer currently receive pertuzumab, so it is not clear whether our results can be applied to patients receiving pertuzumab. However, considering that favorable prognosis in patients with axillary pCR is consistent across different chemotherapy regimens [18,19,20,21,22,23], it is likely that our results will be applicable even to those who receive pertuzumab for HER2-positive disease. Another limitation was the short follow-up period. The median follow-up time after the primary surgery was 64 months. It is important to follow the patients for a longer period to examine the long-term outcomes of patients with axillary pCR.

## 4. Materials and Methods

### 4.1. Patients

We enrolled patients with stage IIA–IIIC breast cancer who underwent surgery at the Breast Oncology Center, Cancer Institute Hospital, Japanese Foundation for Cancer Research, Tokyo, between January 2007 and December 2012. Patients with clinically node-negative disease, bilateral breast cancer, those who received preoperative hormone therapy, and male patients were excluded. Of the 780 patients with cytologically proven node-positive disease, 596 received NAC. The data of the 596 patients are shown in Appendix A. For the control as patients with axillary pN- without NAC, we enrolled patients with pT1-4pN0M0 breast cancer who underwent surgery without NAC at our institution during the same time period (Figure 1). The data of patients for the control are shown in Appendix A. The Ethics Committees of Cancer Institute Hospital (# 2018-1100) approved the study protocol on 19 September 2018.

### 4.2. Definition of Clinical LN Status at Diagnosis

All patients underwent preoperative LN evaluation by palpation and ultrasonography. LN metastasis was confirmed using aspiration cytology when ultrasonography suspected LN metastasis. LN metastasis was suspected by ultrasonography in case of hypoechoic round shape, focally thickened cortex, or absent fatty hilum [35,36].

### 4.3. NAC

Regimens for NAC were based on guidelines provided by the Japanese Breast Cancer Society [37]. The regimens included anthracycline-based or anthracycline followed by taxane chemotherapy. Anthracycline-based regimens inclued 4–6 cycles of CAF (cyclophosphamide 500 mg/m^2^, adriamycin 50 mg/m^2^, and fluorouracil 500 mg/m^2^, q3w); AC (adriamycin 60 mg/m^2^ and cyclophosphamide 600 mg/m^2^, q3w); and CEF (cyclophosphamide 500 mg/m^2^, epirubicin 100 mg/m^2^, and fluorouracil 500 mg/m^2^, q3w). Taxane regimens included 12 cycles of weekly paclitaxel at 80 mg/m^2^ or four cycles of tri-weekly docetaxel at 75 mg/m^2^. Trastuzumab was administered together with and after taxane totally for one year in HER2-positive patients.

### 4.4. Adjuvant Therapy

Adjuvant therapy was based on the guidelines provided by the Japanese Breast Cancer Society [37]. Anthracycline and/or taxane regimens were administered depending on the risk factors, such as tumor size, nodal involvement, hormone receptor status, HER2 status, NG, and Ki-67 status, if not administered at NAC. Endocrine and anti-HER2 therapy were based on hormone receptor and HER2 status, respectively. Post-mastectomy radiotherapy was administered to patients with ≥ 4 positive LNs, 1–3 positive LNs with extensive LI, internal mammary and/or supraclavicular LN metastasis, or inflammatory breast cancer.

### 4.5. Nodal Surgery and LN Pathology

All patients with cytologically proven node-positive breast cancer patients at baseline underwent LN dissection, even without lymphadenopathy after NAC. Patients who achieved axillary pCR were confirmed to be negative for metastasis in all dissected LNs. In patients with axillary pN- without NAC, LN negativity was confirmed by SLNB with radioisotope and/or dye methods.

### 4.6. Definition of Breast-pCR

Pathological tumor response was based on the general rules for clinical and pathological recording of breast cancer (18th edition) [38]. In our study, breast-pCR was defined as no residual invasive cancer cells in the breast regardless of in situ lesions.

### 4.7. Immunohistochemical (IHC) Analysis

Samples were considered positive for ER and progesterone receptor if staining of the tumor cell nuclei was observed in ≥ 10% of the cancer cells. HER2 protein expression was indicated by IHC scores 0, 1+, 2+ and 3+. Samples with IHC score 2+ were further analyzed by in situ hybridization to identify gene amplification. HER2 positivity was defined as HER2 IHC score 3+ or HER2 gene amplification according to the ASCO/CAP guidelines [39].

### 4.8. Follow-Up Data

Follow-up data until 31 October 2018 were collected using the institutional database. No patient was lost to follow-up during the study period. We retrospectively reviewed the clinicopathological characteristics (including age, menopausal status, clinical T stage, clinical N stage, clinical stage, hormone receptor status, HER2 status, and LI status of the operative specimens), treatment modality (surgery, chemotherapy, endocrine therapy, trastuzumab, and radiotherapy), and DDFS and OS. TNM classification was based on the Union for International Cancer Control Staging System (eighth edition) [40]. T described the size of the primary tumor and whether it had invaded nearby tissue. N described regional lymph nodes that were involved. M described distant metastasis. DDFS was defined as the period from the day of primary surgery until the day of diagnosis of distant metastasis or death from any cause. OS was defined as the period from the day of primary surgery until the day of death from any cause.

### 4.9. Propensity Score Matching

Propensity score matching can be used to adjust for baseline characteristics and reduce the effect of selection bias [34,41,42]. Each patient in the study was assessed by a score calculated by potential confounders, and the two cohorts were matched based on these scores. The comparison of outcomes between the two groups (patients with axillary pCR and axillary node-negative disease without NAC) can be fair and avoid bias to some extent. The variables for propensity score matching were selected as follows: menopausal status, cT, ER status, HER2 status, NG. We used a ratio of 1:1 for nearest neighbor matching within 0.2 standard deviations of the logit of the propensity score.

### 4.10. Statistical Analysis

Independent *t*-tests and χ^2^ tests were used to analyze the differences in clinicopathological characteristics between the two groups. The Kaplan–Meier method was used to determine DDFS and OS, and survival curves were analyzed using the log-rank test. All *p* values were two tailed, and *p* < 0.05 was considered statistically significant. All statistical analysis were performed with EZR (Saitama Medical Center, Jichi Medical, Saitama, Japan), which is a graphical user interface for R (The R Foundation for Statistical Computing, Vienna, Austria) [43].

## 5. Conclusions

We showed that axillary pCR after NAC was associated with improved prognosis in patients with cytologically proven axillary node-positive disease. Furthermore, we showed that patients with axillary pCR and matched pairs with axillary pN- without NAC had comparable DDFS and OS. These results will be useful for patients and oncologists when considering the intensity of follow-up and adjuvant therapy in patients with clinically node-positive disease.

## Figures and Tables

**Figure 1 cancers-12-02633-f001:**
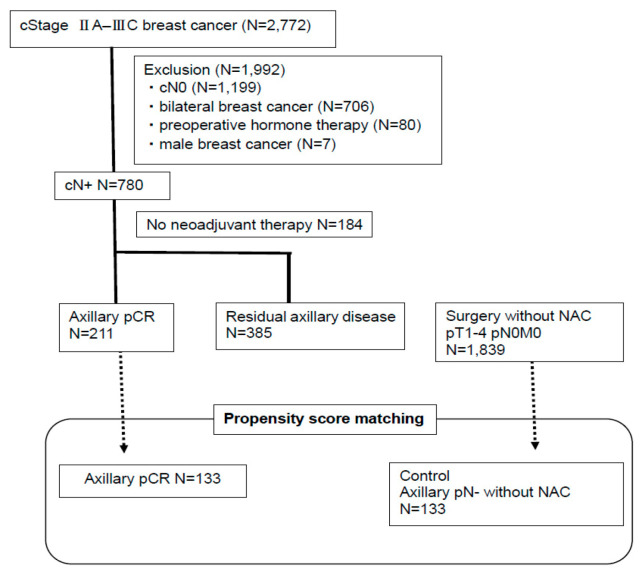
Flow diagram of the study.

**Figure 2 cancers-12-02633-f002:**
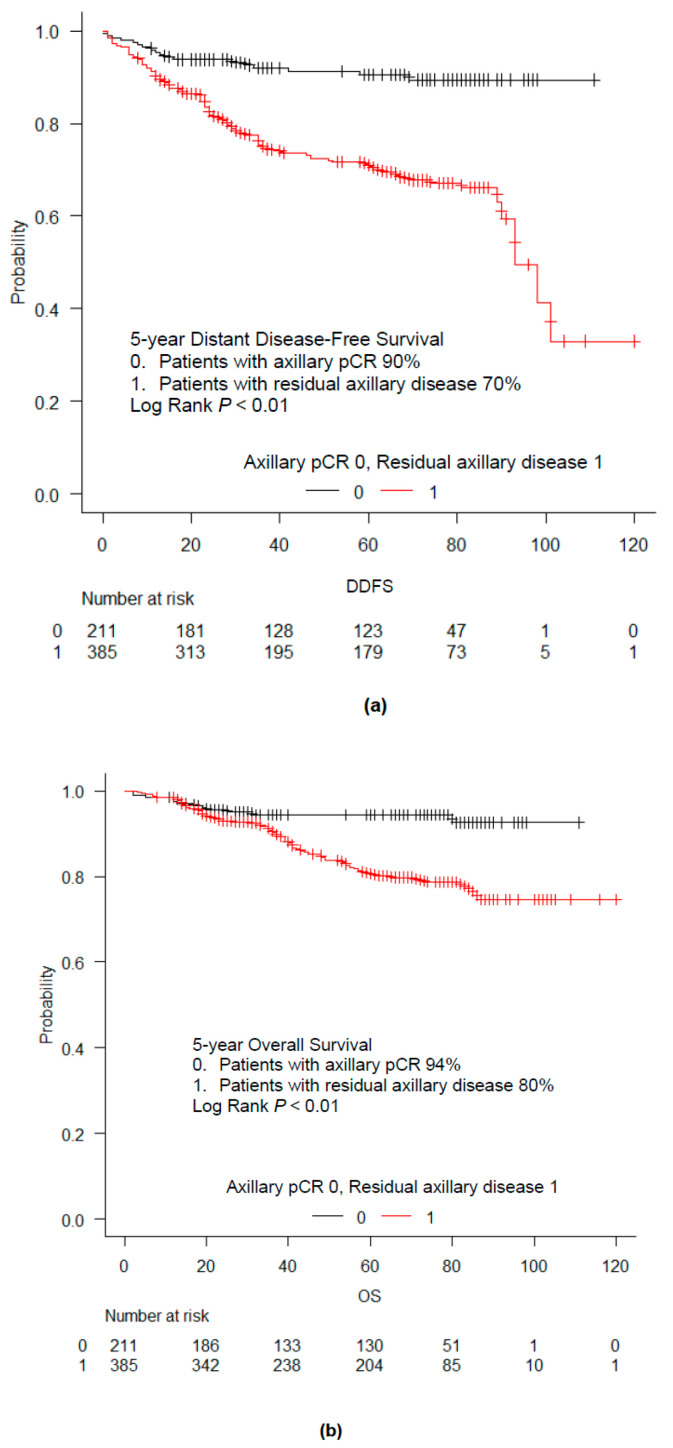
Survival outcomes according to pathological LN response to NAC. Kaplan–Meier curves for distant disease-free survival (DDFS) (**a**) and overall survival (OS) (**b**) in all patients with cytologically proven node-positive breast cancer after NAC. The 5-year DDFS and OS rates were 90% and 94%, respectively, in patients with axillary pCR; 70% and 80%, respectively, in patients with residual axillary disease (*p* < 0.01 for both DDFS and OS). DDFS—distant disease-free survival, NAC—neoadjuvant chemotherapy, OS—overall survival, pCR—pathological complete response.

**Figure 3 cancers-12-02633-f003:**
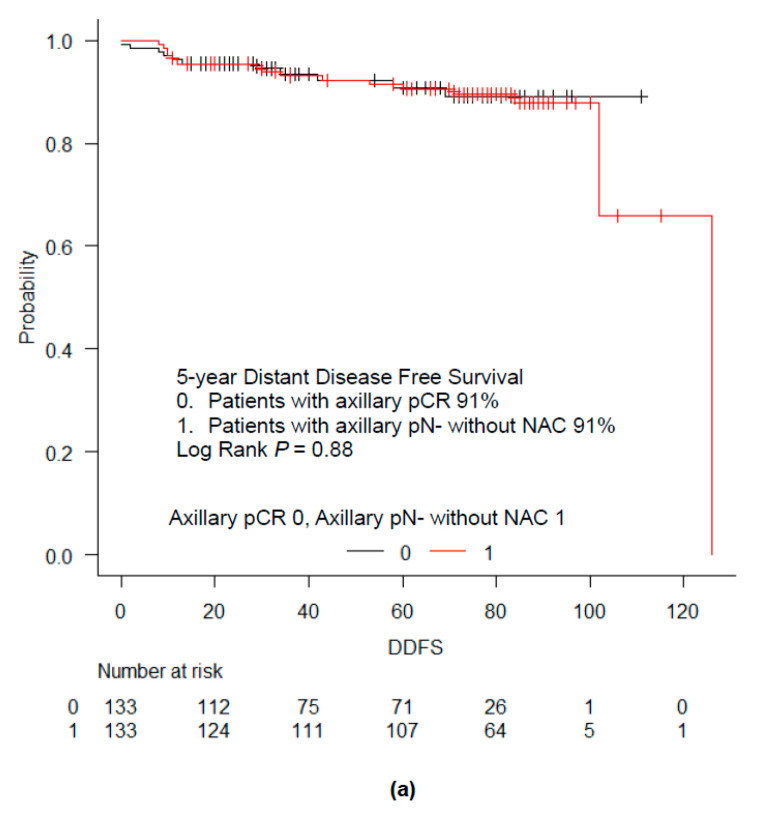
Survival outcomes of patients with axillary pCR and matched pairs with axillary pN- without NAC. Kaplan–Meier curves for DDFS (**a**) and OS (**b**) in patients with cytologically proven node-positive breast cancer with axillary pCR after NAC and matched pairs with axillary pN- without NAC. The log-rank test showed that there was no significant difference in DDFS and OS between patients with axillary pCR and matched pairs with axillary pN- without NAC (*p* = 0.88 for DDFS; *p* = 0.67 for OS).

**Table 1 cancers-12-02633-t001:** Clinicopathological characteristics of patients after neoadjuvant chemotherapy (NAC).

Characteristics	Patients with Axillary pCR (*n* = 211)	Patients with Residual Axillary Disease (*n* = 385)	*p*-Value
Age, years (mean ± SD)	50.7 ± 10.9	51.1 ± 10.8	0.65
Menopausal status			
Pre-	106 (50%)	195 (51%)	0.93
Post-	105 (50%)	190 (49%)	
Clinical T stage ^a^			
T1	31 (14.5%)	48 (13%)	0.02
T2	139 (66%)	221 (57%)	
T3	24(11%)	67 (17%)	
T4	16 (8%)	49 (13%)	
TX	1 (0.5%)	0	
Clinical N stage ^a^			
N1	162 (77%)	306 (80%)	0.65
N2	9 (4%)	18 (4%)	
N3	40 (19%)	61 (16%)	
Clinical stage ^a^			
II	137 (65%)	224 (58%)	0.12
III	74 (35%)	161 (42%)	
ER status			
Negative	115 (55%)	70 (18%)	<0.01
Positive	83 (39%)	295 (77%)	
Unknown	13 (6%)	20 (5%)	
HER2 status			
Negative	90 (43%)	288 (75%)	<0.01
Positive	100 (47%)	52 (13%)	
Unknown	21 (10%)	45 (12%)	
NG			
1	47 (22%)	170 (44%)	<0.01
2	58 (28%)	111 (29%)	
3	82 (38%)	56 (15%)	
Unknown	24 (12%)	48 (12%)	
Type of surgery			
Partial mastectomy	86 (40%)	70 (18%)	<0.01
Mastectomy	125 (60%)	315 (82%)	
Breast pCR			
No	120 (57%)	372 (97%)	<0.01
Yes	91 (43%)	13 (3%)	
LI status of the surgical specimens			
Negative	179 (85%)	211 (55%)	<0.01
Positive	32 (15%)	174 (45%)	
Pre and Postoperative Chemotherapy			
Anthracycline	6 (3%)	11 (3%)	0.29
Taxane	3 (2%)	1 (0/3%)	
Anthracycline followed by taxane	202 (95%)	373 (96.7%)	
Pre and Postoperative Trastuzumab for HER2-positive disease			
Yes	95 (95%)	48 (92%)	0.90
No	5 (5%)	4 (8%)	
Hormone therapy after surgery for ER-positive disease			
Yes	81 (98%)	289 (98%)	0.90
No	2 (2%)	6 (2%)	
Radiotherapy after surgery			
Yes	141 (67%)	281 (73%)	0.13
No	70 (33%)	104 (27%)	

^a^ TNM classification is shown based on the eighth edition of the Union for International Cancer Control staging system. The bar indicates values that are statistically significant (*p* < 0.05). ER—estrogen receptor; HER2—human epidermal growth factor receptor 2; LI—lymphatic invasion; NAC—neoadjuvant chemotherapy; NG—nuclear grade; SD—standard deviation; pCR-—pathological complete response.

**Table 2 cancers-12-02633-t002:** Multivariate analysis of prognostic factors related to DDFS in patients with NAC.

Characteristics	Hazard Ratio	95%CI	*p*-Value
Pathological lymph node status after NAC			
Axillary pCR	1		
Residual axillary disease	4.55	2.48–8.35	<0.01
Clinical stage ^a^			
II	1		
III	2.92	1.96–4.35	<0.01
ER status			
Negative	1		
Positive	0.60	0.37–0.96	0.03
HER2 status			
Negative	1		
Positive	0.92	0.77–1.09	0.35
NG			
1,2	1		
3	1.23	0.96–1.57	0.10

^a^ TNM classification is shown based on the eighth edition of the Union for International Cancer Control staging system. The bar indicates values that are statistically significant (*p* < 0.05). CI—confidence interval; DDFS—distant disease-free survival; ER—estrogen receptor; HER2—human epidermal growth factor receptor 2; NAC—neoadjuvant chemotherapy; NG—nuclear grade; pCR—pathological complete response.

**Table 3 cancers-12-02633-t003:** Factors associated with axillary pCR using multivariate logistic regression analysis.

Characteristics	Odds Ratio	95%CI	*p*-Value
Menopausal status			
Pre-	1		
Post-	0.63	0.37–1.04	0.07
Clinical T ^a^			
1	1		
2–4	0.91	0.40–1.29	0.79
Clinical N ^a^			
1	1		
2,3	0.72	0.40–1.29	0.27
ER status			
Negative	1		
Positive	0.40	0.23–0.70	<0.01
HER2 status			
Negative	1		
Positive	1.49	1.24–1.78	<0.01
NG			
1,2	1		
3	1.51	1.15–2.00	<0.01
Breast-pCR			
No	1		
Yes	3.81	2.52–5.76	<0.01

^a^ TNM classification is shown based on the eighth edition of the Union for International Cancer Control staging system. The bar indicates values that are statistically significant (*p* < 0.05). CI—confidence interval; ER—estrogen receptor; HER2—human epidermal growth factor receptor 2; NG—nuclear grade; pCR—pathological complete response.

**Table 4 cancers-12-02633-t004:** Baseline characteristics at diagnosis of patients with axillary pCR and axillary pN- without NAC, using propensity score matching.

Characteristics	Axillary pCR (*n* = 133)	Axillary pN- without NAC (*n* = 133)	*p*-Value
Menopausal status			
Pre-	68	69	1.00
Post-	65	64	
Clinical T ^a^			
1	23	27	0.78
2	102	99	
3	7	5	
4	1	2	
ER status			
Negative	71	67	0.62
Positive	62	66	
HER2 status			
Negative	73	81	0.38
Positive	60	52	
NG			
1	31	33	0.18
2	35	47	
3	67	53	
Type of surgery			
Partial mastectomy	61	70	0.33
Mastectomy	72	63	
Pre and Postoperative Chemotherapy			
No	0	61	<0.01
Anthracycline	5	63	
Taxane	2	3	
Anthracycline followed by taxane	126	3	
Others	0	3	
Pre and Postoperative Trastuzumab for HER2-positive disease			
Yes	58	38	0.02
No	75	95	
Hormone therapy after surgery for ER-positive disease			
Yes	61	52	0.39
No	72	81	
Radiotherapy after surgery			
Yes	88	38	<0.01
No	45	95	

^a^ TNM classification is shown based on the eighth edition of the Union for International Cancer Control staging system. The bar indicates values that are statistically significant (*p* < 0.05). ER—estrogen receptor; HER2—human epidermal growth factor receptor 2; NG—nuclear grade; pCR—pathological complete response.

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
