# Peer review of "Survival in Cytologically Proven Node-Positive Breast Cancer Patients with Nodal Pathological Complete Response after Neoadjuvant Chemotherapy"

_cancers, 2020, doi:10.3390/cancers12092633_

Round 1
Reviewer 1 Report
Although the authors of this study try to make an interesting point in the comparison of different patient groups, the description of the groups is lacking precision and the text requires extensive editing of language and style. definitions are not given, abbreviations not explained. One can see that the topic could be interesting, but it is so hard to identify the arguments and points the authors are trying to make. I would like to read an improved version of this text very much but in its present form it is not fit for publication.
Author Response
Response: We have revised the manuscript with fewer abbreviations and clearer definitions of each term.
We have added the description of the control cohort in the Material and Methods section (Line 236, 237) and the definition of patients with axillary pN- without NAC in the Material and Methods section (Line 267, 268).
We have had this manuscript edited by the native English writer using the English editing service.
Reviewer 2 Report
Thank you for the opportunity to review this very interesting paper. This is a retrospective observational study involving two arms: 1) patients with cytologically proven node-positive breast cancer who underwent surgery after NAC, 2) and those with LN-negative disease without NAC for control between January 2007 and December 2012. The aim was to compare the survival outcome between the two groups. Overall, the paper is well written and nicely presented.
Minor editing suggestions:
1. Too many abbreviations, which makes it difficult to read and require tracking down the location of where the abbreviations are explained. I may have missed location of some abbreviations used. For example, I could not find explanation for ypN+. Suggest reducing the abbreviations and not combining abbreviations. For example, LN meaning lymph node is commonly known and can leave. Spell out PSM. Make sure all abbreviations have explanation at the beginning.
2. The order of the sections are not consistent with journal instructions. For example, methodology (section 4) is after "Discussion section" or section 3. Did I misunderstood the instructions on the Journal website?
3. The first statement in the conclusion is very important. It should be 2 sentences since 2 separate thoughts and would help to emphasize your findings. 1) Patients with initial biopsy proven metastatic node from breast cancer who had complete pathological response after NAC had improve overall survival compared to those who did not have complete nodal response after NAC, 2) Patients with initial biopsy proven metastatic node from breast cancer who received NAC and had complete pathological response had similar overall survival to those patients with no nodal disease at initial staging. Is my understanding correct?
Author Response
- Response: We have revised the manuscript with fewer abbreviations and clearer definitions of each term. We have changed “LN-pCR” to “axillary pCR”, “ypN+” to “residual axillary disease”, and “LN-negative” to “axillary pN-” to avoid combining abbreviations. We have spelled out PSM (propensity score matching). We have reviewed all abbreviations and explanations to make them clearer.
- Response: According to the Instructions for Authors, we have put the “Material and Methods” section after “Discussion section” and before “Conclusion”.
- Response: Thank you for raising the important point. We agree and have revised the statement as you indicated into two sentences (Line 311-313).
Round 2
Reviewer 1 Report
The changes make for an easier read, the manuscript is acceptable.